# Effectiveness of BBIBP-CorV vaccine against severe outcomes of COVID-19 in Abu Dhabi, United Arab Emirates

Nawal Al Kaabi[1,2,20 ✉], Abderrahim Oulhaj[2,3,4,20], Subhashini Ganesan [5,6], Farida Ismail Al Hosani[7], Omer Najim[8], Halah Ibrahim [2], Juan Acuna[2,4], Ahmed R. Alsuwaidi[3], Ashraf M. Kamour[1], Ashraf Alzaabi[3], Badreyya Ahmed Al Shehhi[7], Habiba Al Safar [9,10,11], Salah Eldin Hussein[1], Jehad Saleh Abdalla[1], Dalal Saeed Naser Al Mansoori[12], Ahmed Abdul Kareem Al Hammadi[12], Mohammed A. Amari[1], Ahmed Khamis Al Romaithi[1], Stefan Weber[13], Santosh Elavalli[5], Islam Eltantawy[6], Noura Khamis Alghaithi[14], Jumana Nafiz Al Azazi[15,16], Stephen Geoffrey Holt[17], Mohamed Mostafa[18], Rabih Halwani [19], Hanif Khalak[5], Wael Elamin [5], Rami Beiram[3] & Walid Zaher[2,3,5,6]

The effectiveness of the inactivated BBIBP-CorV vaccine against severe COVID-19 outcomes (hospitalization, critical care admission and death due to COVID-19) and its long-term effectiveness have not been well characterized among the general population. We conducted a retrospective cohort study using electronic health records of 3,147,869 adults, of which 1,099,886 vaccinated individuals were matched, in a 1:1 ratio to 1,099,886 unvaccinated persons. A Cox-proportional hazard model with time varying coefficients was used to assess the vaccine effectiveness adjusting for age, sex, comorbidity, ethnicity, and the calendar month of entry into the study. Our analysis showed that the effectiveness was 79.6% (95% CI, 77.7 to 81.3) against hospitalization, 86% (95% CI, 82.2 to 89.0) against critical care admission, and 84.1% (95% CI, 70.8 to 91.3) against death due to COVID-19. The effectiveness against these severe outcomes declined over time indicating the need for booster doses to increase protection against severe COVID-19 outcomes.

[1] Sheikh Khalifa Medical City SEHA, Abu Dhabi, UAE. [2] College of Medicine and Health Sciences, Khalifa University, Abu Dhabi, UAE. [3] College of Medicine and Health Sciences, United Arab Emirates University, Al Ain, UAE. [4] Research and Data Intelligence Support Center, Khalifa University, Abu Dhabi, UAE. [5] G42 Healthcare, Masdar City, Abu Dhabi, UAE. [6] IROS (Insights Research Organization & Solutions), Abu Dhabi, UAE. [7] Abu Dhabi Public Health Center – ADPHC, Abu Dhabi, UAE. [8] Department of Health (DOH), Abu Dhabi, UAE. [9] Khalifa University of Science and Technology, Abu Dhabi, UAE. [10] College of Engineering, Khalifa University of Science and Technology, Abu Dhabi, UAE. [11] Emirates Bio-Research Center, Ministry of Interior, Abu Dhabi, UAE. [12] Tawam hospital SEHA, Al Ain, UAE. [13] Reference Laboratory for Infectious Diseases (RLID) and Union 71, Abu Dhabi, UAE. [14] Ambulatory Health Services- SEHA, Abu Dhabi, UAE. [15] Abu Dhabi Health Services Company- SEHA, Abu Dhabi, UAE. [16] Abu Dhabi Executive Office, Abu Dhabi, UAE. [17] SEHA Kidney Care, SEHA, Abu Dhabi Health Services Company, Abu Dhabi, UAE. [18] PDC -CRO, Clinical research organization, Dubai, UAE. [19] College of Medicine, University of Sharjah, Sharjah, UAE. [20] These authors contributed equally: Nawal Al Kaabi, Abderrahim Oulhaj. ✉email: alkaabin971@gmail.com

The coronavirus disease 2019 (COVID-19) pandemic, caused by the SARS-CoV-2 virus, as of May 2022 has caused over 6.2 million deaths globally[1]. Vaccination programs against COVID-19 infection have been enthusiastically embraced by most countries, with more than 11.6 billion vaccination doses administered to date worldwide[2]. Although several COVID-19 vaccines are in large-scale administration throughout the world, studies are required to understand real world vaccine effectiveness, which is likely to differ from vaccine efficacy reported in controlled clinical trials. Hospital and community-based studies reflecting real world evidence of COVID-19 vaccine effectiveness have been published on several vaccines[3–5]. In England, a test negative case control study of an elderly population demonstrated that a single dose of either the BNT162b2 mRNA (Pfizer BioNTech) or ChAdOx1 (AstraZeneca) vaccine was around 80% effective in preventing COVID-19 hospital admission[3]. In Scotland, an interim study conducted after one dose of BNT162b2 mRNA or ChAdOx1 vaccine noted that the vaccine effectiveness in preventing hospital admission was 91% and 88%, respectively[4]. In Israel, the BNT162b2 mRNA vaccine was reported to be 97% effective in preventing hospital and critical care admissions for a period of 4 months[5].

The United Arab Emirates (UAE), in collaboration with China National Biotec Group Company Limited, Beijing, was the first country to commence a phase III trial for an inactivated SARS-CoV-2 BBIBP-CorV (Sinopharm) vaccine in one of the largest COVID-19 vaccine trials conducted to date[6]. The BBIBP-CorV vaccine was developed from the HB02 strain of SARS-CoV-2 virus, which was purified and passaged into Vero cells to generate the stock for vaccine production. The virus was then inactivated by beta-propiolactone and mixed with an aluminum-based adjuvant[7]. The phase III trial concluded that the recommended two doses of BBIBP-CorV vaccine, given at a three-week interval, was 78% effective in preventing symptomatic COVID-19 infections[8]. Based on these results, the UAE government authorized emergency approval of the BBIBP-CorV vaccine for frontline workers in September 2020 and for public use in December 2020[9]. The World Health Organization also approved the vaccine for emergency use globally[10]. The UAE has one of the highest COVID-19 vaccination rates worldwide, with over 24.6 million doses administered, and more than 97% of the adult population fully vaccinated against SARS-CoV-2 infection[11].

A number of studies are now reporting a reduction in vaccine effectiveness against COVID-19 infections over time[12,13]. A retrospective cohort study conducted in the United States noted that the effectiveness against infection declined from 88% during the first month after complete vaccination with the BNT162b2 mRNA vaccine to 47% at five months post-vaccination[12]. Another study in Qatar showed the effectiveness of the same vaccine against SARS-CoV-2 infections declined gradually, with an accelerated decline after the fourth month, resulting in approximately 20% protection at five to seven months post-vaccination. Nevertheless, the effectiveness remained at nearly 96% in preventing hospitalization and death at six months post-vaccination[13]. A recent study from the UAE on the inactivated BBIBP-CorV vaccine showed that the effectiveness against severe outcomes of COVID-19 was 80%, 92% and 97% against hospitalization, critical care admission, and death respectively[14]. Another study from Morocco on long-term effectiveness of the inactivated BBIBP-CorV vaccine showed that the effectiveness declined from 88% to 64% in six months post-vaccination[15]. The waning effectiveness of vaccines might be due to the fact that the anti-spike protein and other neutralizing antibody titers reduce over time after vaccination, but the emergence of newer variants may also contribute to the reduction in vaccine effectiveness[16,17].

This study was conducted to evaluate the real-world effectiveness of two doses of the inactivated BBIBP-CorV vaccine in preventing severe COVID-19 outcomes, including hospital admissions, admissions to critical care, and COVID-19 related deaths over a period of three months in the emirate of Abu Dhabi, UAE, and to assess the decline in the vaccine effectiveness over time against each of the studied COVID-19 outcomes.

## Results

**Study population and matching**. A total of 1,832,583 vaccinated individuals who received two doses of inactivated BBIBP-CorV vaccine and completed their second dose prior to July 1st, 2021, and 1,315,286 unvaccinated controls who did not receive any dose of this or any other vaccine prior to July 1st, 2021, were retrieved from the database. Rolling Entry Matching procedure was employed and resulted in matching 1,153,515 vaccinated individuals in a 1:1 ratio to 1,153,515 unvaccinated controls according to age, sex, ethnicity, comorbidities and the date of entry into the study.

Among the 1,153,515 matched vaccinated individuals, 2356 (0.20%) were excluded, along with their matched controls, because they received their second dose less than two weeks after the first dose. Of the remaining matched individuals, 12,848 (5709 vaccinated and 7139 unvaccinated individuals corresponding to 12661 unique matched pairs) had COVID-19 hospitalization prior to their baseline dates. These participants, along with their corresponding matched individuals, were also excluded. Of the remaining matched controls, 38,612 (3.35%) were vaccinated during their follow-up period and were also excluded from the primary analysis, along with their matched vaccinated individuals, resulting in a total of 2,199,772 individuals (1,099,886 vaccinated and 1,099,886 unvaccinated matched controls) used in the primary analysis. In a sensitivity analysis, these unvaccinated controls with post-baseline vaccination were re-included, along with their matched vaccinated individuals, resulting in a total of 2,276,996 individuals (1,138,498 vaccinated and 1,138,498 unvaccinated matched controls) (Fig. 1- Flowchart).

**Demographic and clinical characteristics at baseline**. Table 1 shows the baseline demographic and clinical characteristics of the 2,199,772 matched individuals (vaccinated versus unvaccinated) that were included in the analysis. The median age for vaccinated individuals was 35 years compared to 34 years for their matched non-vaccinated controls. The same proportion of females (36%) was observed in both vaccinated and matched unvaccinated individuals. Vaccinated individuals comprised 3.8% of people older than 60 years old, compared to 5.6% of their matched unvaccinated controls. The distribution of the other demographics and clinical features, including comorbidities, did not differ substantially between the two groups.

**Overall effectiveness**. During the follow-up period of three months, there were 3505 COVID-19 related hospitalizations (2889 in unvaccinated compared to 616 in vaccinated), 653 critical care admissions (574 in unvaccinated compared to 79 in vaccinated), and 99 deaths (87 in unvaccinated versus 12 in vaccinated) documented. Cumulative incidence functions, along with the incidence rate of the three COVID-19 outcomes, are displayed in Fig. 2. Furthermore, the frequency of COVID-19 outcomes according to different risk factors and vaccine status is provided in supplementary material (Supplementary Table 1).

Adjusted hazard ratios, with their corresponding 95% CIs using the Cox proportional-hazards model, were estimated for the three COVID-19 outcomes (Supplementary Table 2). The

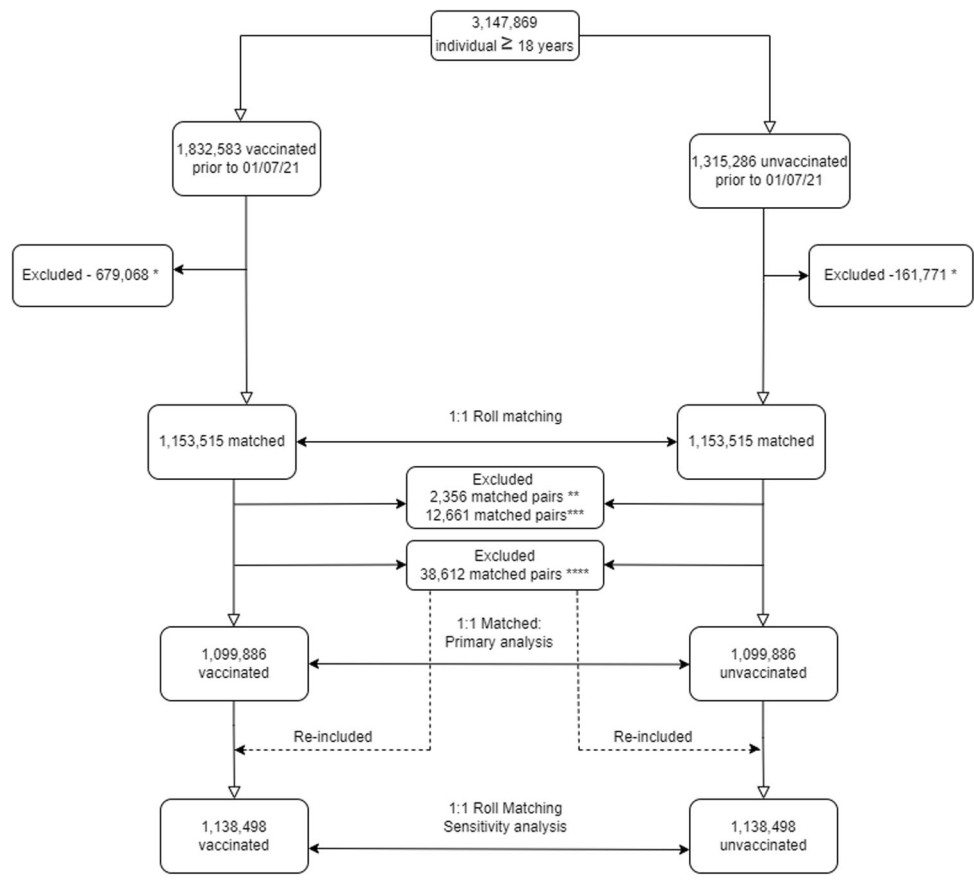

**Fig. 1 Flowchart of study population and matching process.** *Were not matched. **Vaccinated individuals with the interval between 1st and 2nd dose less than 2 weeks. Their corresponding matched unvaccinated individuals were also excluded. ***Individuals (Vaccinated & Unvaccinated) who had hospital admission prior to their baseline dates. Their corresponding matched individuals (Vaccinated & Unvaccinated) were also excluded. ****Unvaccinated individuals who received their first dose of the vaccine during their follow-up period (i.e. after their baseline date). Their corresponding matched vaccinated individuals were also excluded.

estimated adjusted vaccine effectiveness at three months post complete vaccination was 79.6% (95% CI, 77.7–81.3, $P < 0.001$) against COVID-19 related hospitalization, 86% (95% CI, 82.2–89.0, $P < 0.001$) against critical care admission, and 84.1% (95% CI, 70.8–91.3, $P < 0.001$) against death. The results of the sensitivity analysis, which includes unvaccinated controls who had at least one dose of the vaccine during their follow-up period showed similar results for effectiveness against hospitalization (77.4%, 95% CI 75.4–79.3, $P < 0.001$), critical care admission (84.3%, 95% CI 80.3–87.5, $P < 0.001$) and death (84.2%, 95% CI 70.9–91.4, $P < 0.001$)). Complete details of the sensitivity analysis are presented in the supplementary material (Supplementary Tables 5, 6 and 7).

**Effectiveness according to different risk factors**. Adjusted vaccine effectiveness at three months post-vaccination was also calculated according to different risk factors. This was done by adding an interaction term between vaccination status and each of the risk factors when modeling each outcome. For hospital admission, we found significant interactions with age, sex, comorbidity, and the month of study entry. Significant interactions with sex, comorbidity, and the month of study entry were observed for critical care admission, whereas mortality due to COVID-19 had significant interactions with comorbidity and the month of study entry only. Vaccine effectiveness against COVID-19 hospitalization was observed to be higher in females (82.3%, 95% CI 80.0–84.3) compared to males ($p < 0.001$), in individuals less than 60 years old (84.7%, 95% CI 82.7–86.4) compared to

those above 60 years of age ($p < 0.001$). Vaccine effectiveness among individuals with comorbidities was higher (83.5%, 95% CI 81.6–85.3) compared to individuals without any associated comorbid conditions ($p < 0.001$). We also observed higher vaccine effectiveness during the months of October through December 2020 (97.5%, 95% CI 95.9–98.4) compared to the months between January 2021 and July 2021 ($p < 0.001$) (Table 2). Vaccine effectiveness was highest against critical care admission among males than females (89.3%, 95% CI 85–92.5, $P < 0.001$), in those with comorbidities compared to those without comorbidities (88.7%, 95% CI, 84.8–91.6, $P < 0.001$), and during the months of October through December 2020 compared to the period between January 2021 to July 2021 (98.8%, 95% CI, 95.3–99.7, $P < 0.001$) (Table 2). Vaccine effectiveness against death was modified by comorbidity and the month of study entry with highest level observed for patients with comorbidity (97.1%, 95% CI, 93.5–98.7, $P < 0.001$), and during the months of October through December 2020 (99.9%, 95% CI, 99.8–100, $P < 0.001$) (Table 2). The sensitivity analysis yielded similar results (Supplementary Tables 8 and 9).

**The waning of effectiveness**. To investigate the waning in vaccine effectiveness over time against each of the COVID-19 outcomes, we assessed the vaccine effectiveness at every month post-vaccination over an extended follow-up period of about twelve months up to 30th of September 2021. The vaccine effectiveness against COVID-19 hospitalizations decreased with increased duration of follow-up from the date of complete vaccination

**Table 1 Demographic and Clinical Characteristics of the study participants at baseline.**

| Characteristics | Overall (n=2,199,772) | Non-vaccinated (n = 1,099,886) | Vaccinated (n = 1,099,886) |
|---|---|---|---|
| Median age (IQR) | 35 (28–43) | 34 (28–43) | 35 (29–43) |
| Age group n (%) | | | |
| Less than 40 years | 1,447,391 (66) | 728,893 (66) | 718,498 (65) |
| 40–60 years | 648,813 (29) | 309,459 (28) | 339,354 (31) |
| More than 60 years | 103,568 (4.7) | 61,534 (5.6) | 42,034 (3.8) |
| Sex n (%) | | | |
| Female | 798,999 (36) | 397,981 (36) | 401,018 (36) |
| Male | 1,400,773 (64) | 701,905 (64) | 698,868 (64) |
| Ethnicity n (%) | | | |
| Arab | 760,211 (35) | 375,887 (34) | 384,324 (35) |
| Asian | 1,209,118 (55) | 605,694 (55) | 603,424 (55) |
| Other | 230,443 (10) | 118,305 (11) | 112,138 (10) |
| Comorbidities n (%) | | | |
| Asthma | 25,581 (1.2) | 11,412 (1.0) | 14,169 (1.3) |
| Chronic kidney disease | 17,321 (0.8) | 8055 (0.7) | 9266 (0.8) |
| Diabetes | 83,583 (3.8) | 40,112 (3.6) | 43,471 (4.0) |
| Heart disease | 13,934 (0.6) | 6433 (0.6) | 7501 (0.7) |
| Hypertension | 47,841 (2.2) | 22,812 (2.1) | 25,029 (2.3) |
| Immunodeficiencies | 2641 (0.1) | 1271 (0.1) | 1370 (0.1) |
| Neoplasms | 19,356 (0.9) | 9525 (0.9) | 9831 (0.9) |
| Respiratory diseases | 3542 (0.2) | 1680 (0.2) | 1862 (0.2) |
| History of transplantation | 1308 (<0.1) | 656 (<0.1) | 652 (<0.1) |
| Comorbidities n (%) | | | |
| No comorbidity | 2,076,268 (94) | 1,044,356 (95) | 1,031,912 (94) |
| One comorbidity | 70,096 (3.2) | 29,750 (2.7) | 40,346 (3.7) |
| 2 or more comorbidities | 53,408 (2.4) | 25,780 (2.3) | 27,628 (2.5) |
| Months of observation n (%) | | | |
| Oct–Dec 2020 | 131,850 (6.0) | 65,925 (6.0) | 65,925 (6.0) |
| Jan–April 2021 | 1,774,370 (81) | 887,185 (81) | 887,185 (81) |
| May–July 2021 | 293,552 (13) | 146,776 (13) | 146,776 (13) |
| Month of observation n (%) | | | |
| October 2020 | 1238 (<0.1) | 619 (<0.1) | 619 (<0.1) |
| November 2020 | 64,340 (2.9) | 32,170 (2.9) | 32,170 (2.9) |
| December 2020 | 66,272 (3.0) | 33,136 (3.0) | 33,136 (3.0) |
| January 2021 | 247,616 (11) | 123,808 (11) | 123,808 (11) |
| February 2021 | 801,792 (36) | 400,896 (36) | 400,896 (36) |
| March 2021 | 392,396 (18) | 196,198 (18) | 196,198 (18) |
| April 2021 | 115,710 (5.3) | 57,855 (5.3) | 57,855 (5.3) |
| May 2021 | 216,856 (9.9) | 108,428 (9.9) | 108,428 (9.9) |
| June 2021 | 192,334 (8.7) | 96,167 (8.7) | 96,167 (8.7) |
| July 2021 | 101,218 (4.6) | 50,609 (4.6) | 50,609 (4.6) |

Details of ethnicities included in the other group are listed in Supplementary Table 11.

(14 days after second dose). The effectiveness against COVID-19 hospitalization declined from 82.8% (95% CI, 80.5–84.8) at two months after complete vaccination to 62.1% (95% CI 60.2–64.0) at 6 months after complete vaccination. While there was gradual decline in the vaccine effectiveness against COVID-19 hospitalization from two to six months post-vaccination, no further decline was noticed from seven to twelve months post-vaccination. Waning of protection was also observed in critical care admission; however, the decline was smaller in comparison to the decline in effectiveness against hospitalization, with an observed effectiveness of 85.7% (95% CI, 80.3–89.6) at two months after complete vaccination to 72.8% (95% CI, 68.8–76.3) at six months post complete vaccination, without further decline from seven to twelve months post-vaccination. The vaccine effectiveness against mortality due to COVID-19 remained above 80% throughout and did not show significant decline over the 12-month follow-up period (Fig. 3).

## Discussion

This study provides estimates of the effectiveness of the inactivated BBIBP-CorV (Sinopharm) vaccine for the prevention of COVID-19-related hospitalization, critical care admission, and death in a large population in the emirate of Abu Dhabi. Our study found that two doses of inactivated BBIBP-CorV vaccine were 79.6%, 86%, and 84.1% effective in preventing COVID-19-related hospitalization, critical care admission, and death, respectively. The effectiveness against critical care admissions and death was lower in our study compared to a recently published study in the UAE on BBIBP-CorV vaccine, which included a smaller sample size and a shorter follow-up duration. Further, the inclusion of only patients who had COVID-19 infection may have over-estimated the vaccine effectiveness[14].

Our findings showed lower effectiveness than those reported in the secondary analysis of the phase III trial, which showed 100% efficacy against severe COVID-19 cases, which included a composite of severe cases and death[8]. This might be due to the fact that our study represents real world data including high risk populations, with a longer follow-up period, compared to median follow-up of 77 days in the phase III trial. The emergence of newer variants of concern of SARS-CoV-2 virus during this study period, which have been shown to escape vaccine-induced antibodies, may have also contributed to the lower effectiveness[17].

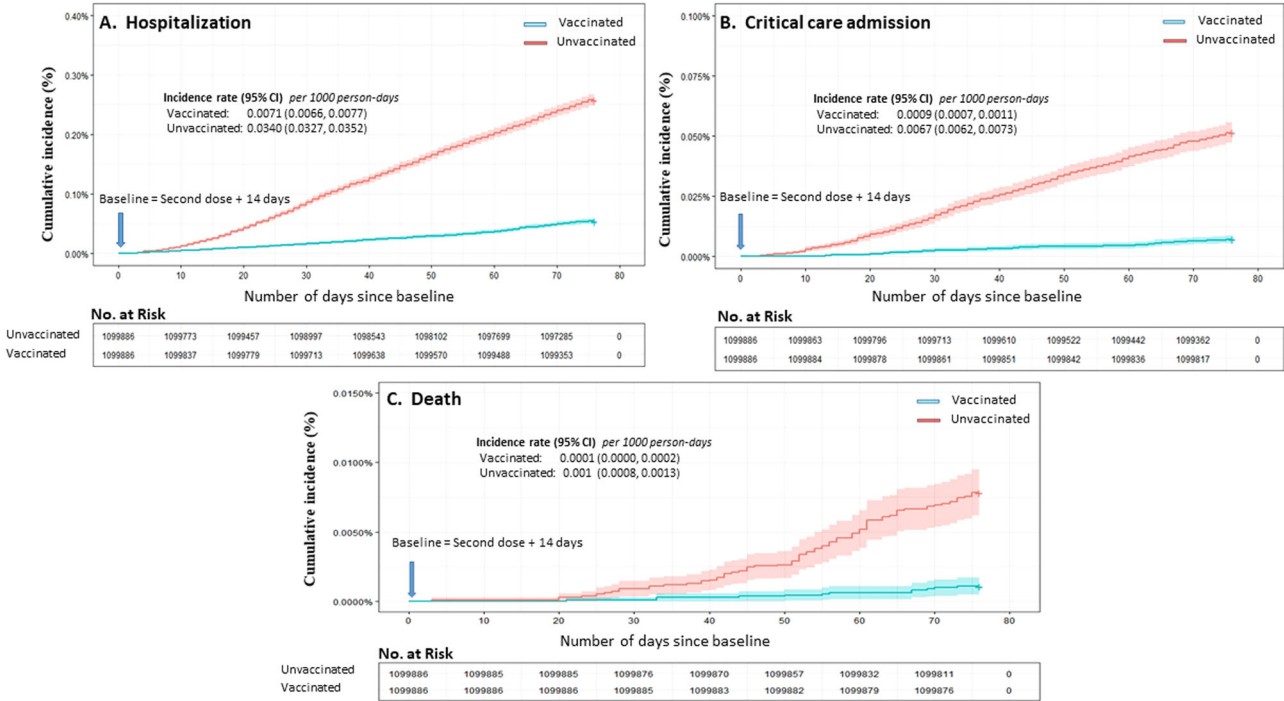

**Fig. 2 Cumulative incidence of severe outcomes of COVID-19.** The blue curve represents vaccinated subjects and the red curve represents unvaccinated individuals. The error bands represent the 95% confidence intervals. The No. at Risk table in each plot represents the number of participants at risk for the outcome at given follow-up times. **A** The cumulative incidence curve of hospitalization. **B** The cumulative incidence curve of critical care admission. **C** The cumulative incidence curve of death.

| Risk factor | Hospitalization | | Critical care admission | | Death | |
|---|---|---|---|---|---|---|
| | Effectiveness (%) (95% CI) | p value | Effectiveness (%) (95% CI) | p value | Effectiveness (%) (95% CI) | p value |
| Months | | **<0.001** | | **<0.001** | | **<0.001** |
| Oct–Dec 2020 (wild-type virus) | 97.5 (95.9, 98.4) | | 98.8 (95.3, 99.7) | | 100 (100, 100) | |
| Jan–April 202 (alpha variant) | 75.9 (73.4, 78.1) | | 81.2 (75.6, 85.5) | | 81.9 (66.8, 90.1) | |
| May–July 2021 (delta variant) | 41.8 (22.9, 56.0) | | 60.1 (9.4, 82.4) | | 62.7 (31.7, 79.6) | |
| Comorbidity | | **<0.001** | | **0.004** | | **<0.001** |
| Without any comorbidity | 68.7 (63.9, 72.9) | | 76.4 (64.5, 84.3) | | 60.7 (12.2, 82.4) | |
| With one or more comorbidities | 83.5 (81.6, 85.3) | | 88.7 (84.8, 91.6) | | 97.1 (93.5, 98.7) | |
| Ethnicity | | 0.58 | | 0.65 | | 0.51 |
| Arab | 79.3 (77.1, 81.3) | | 85.5 (80.8, 89.1) | | 85.6 (71.1, 92.8) | |
| Non-Arab | 80.4 (76.7, 83.6) | | 87.1 (79.9, 91.7) | | 76.9 (20.5, 93.3) | |
| Age group | | **<0.001** | | 0.61 | | 0.24 |
| ≤60 years | 84.7 (82.7, 86.4) | | 86.7 (81.8, 90.3) | | 92.3 (67.2, 98.2) | |
| >60 years | 68.4 (63.9, 72.3) | | 85 (78.3, 89.6) | | 80.1 (61.1, 89.8) | |
| Gender | | **0.0003** | | **0.0155** | | 0.24 |
| Female | 82.3 (80, 84.3) | | 80.8 (73.1, 86.2) | | 77 (47.9, 89.8) | |
| Male | 75.5 (72.2, 78.5) | | 89.3 (85, 92.5) | | 88.9 (72.1, 95.6) | |

**Table 2 Effectiveness of the Vaccine in Preventing Severe COVID-19 Outcomes stratified according to different risk factors.**

P values shown are for testing the interaction between each risk factor and vaccine status. Only statistically significant interactions (shown in bold) can be interpreted as significant subgroup differences. P < 0.05 are considered statistically significant.
P values were calculated using two-sided tests with no adjustment for multiple comparisons, and were computed using the Wald test on the interaction term in the Cox proportional hazard model. Non-Arab population includes Asian and other ethnicity groups as listed in Supplementary Table 11. CI Confidence Interval.

Using the Cox proportional hazard model with interaction terms, we analyzed the effectiveness according to age group, sex, comorbidities, ethnicity, and the month of observation. We found that vaccine effectiveness against hospitalization declined with age, as confirmed by reduced vaccine protection in the older age group (≥60 years), when compared to the younger study population. These findings are supported by other studies, which reported an inverse relationship between COVID-19 vaccine effectiveness and age[18–20]. Our reported vaccine effectiveness against hospitalization in the older age group is likely to be

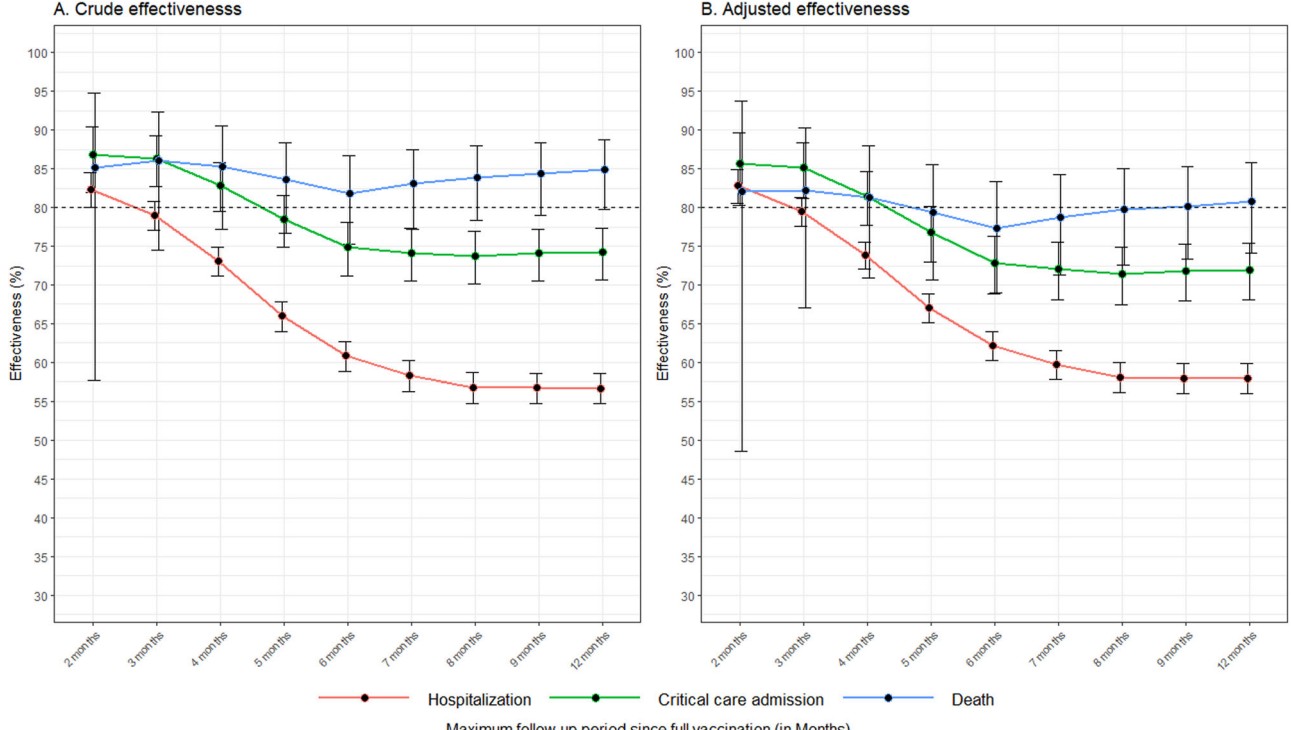

**Fig. 3 Decline in vaccine effectiveness over 12 months of follow up.** Vaccine effectiveness estimated at increasing follow-up periods using the Cox proportional hazard regression model. The red, green and blue curves indicate hospitalization, critical care admission and death respectively. Error bars represent the 95% confidence intervals of the true effectiveness, $n = 2,199,772$ individuals (1,099,886 vaccinated and 1,099,866 unvaccinated matched controls). **A** Effectiveness estimated without adjustment. **B** Effectiveness estimated adjusting for age, sex, presence of comorbidity, ethnicity, and the calendar month of entry into the study.

underestimated as UAE regulations mandated the hospitalization of all patients above the age of 65 years with positive RT-qPCR (Reverse Transcription Quantitative real-time PCR) test for SARS-CoV-2, irrespective of vaccination status as a proactive measure to prevent severe complications in this vulnerable population[21]. Since 40% of hospitalizations in our study occurred in patients ≥60-years of age, this result needs cautious interpretation. The difference between the two age groups in terms of vaccine effectiveness against critical care admission and death were not statistically significant.

With regard to sex, our study showed that the vaccine is more effective in preventing hospitalization in females (82.3%), as compared to males (74.5%), which is supported by another study on inactivated CoronaVac vaccine[22]. Yet, males were more protected against critical care admission. No significant sex differences in mortality were found.

While many studies have reported reduced vaccine effectiveness among people with comorbidities[17,20], our study found augmented vaccine effectiveness against hospitalization, critical care admission and death in people with comorbidities, when compared to those without comorbidities. This might be explained by the high risk of hospitalization among unvaccinated individuals with comorbidities, when compared to the significantly lower risk in unvaccinated individuals without comorbidities. As such, there was greater reduction in the risk after vaccination in individuals with comorbidities compared to those without comorbidities, and thereby increased vaccine effectiveness, in the population with comorbidities.

There were fewer deaths in our population, when compared to other studies. This may be due to the proactive measures taken in Abu Dhabi, including the hospitalization of all high-risk patients,

irrespective of symptom severity or health insurance status. In addition, all patients with COVID-19 infection received standardized treatment in dedicated COVID-19 hospitals.

It is notable that vaccine effectiveness against all COVID-19 outcomes was high during the months of October to December 2020, likely because the emergency use authorization vaccination rollout for frontline workers in the UAE started in September 2020, and this time period preceded the emergence and circulation of variants of concern[9]. As vaccine effectiveness was measured in three-month intervals to control for waning effectiveness over time, the reduction in vaccine effectiveness during the months of January to June 2021 is likely due to the emergence of novel variants of the SARS-CoV-2 virus, some of which had increased transmissibility, infectivity, morbidity, and mortality[17,22]. During this time period, the alpha and delta variants of SARS-CoV-2 were the most prevalent circulating strains in the UAE[23,24].

The twelve months follow-up revealed that the effectiveness of the vaccine started to decline after three months, and that the effectiveness against critical care admissions dropped below 75% after six months. Similar results were reported from another study that observed the long-term effectiveness of BBIBP-CorV vaccine to decline from 88% to 64% in six months[15]. Other studies also demonstrated a gradual waning of the immune response and vaccine effectiveness within months after vaccination with the BNT162b2 vaccine[12,16,25]. This decline in effectiveness over time is likely explained by the viral evasion of vaccine induced immune response through antigenic changes in newer variants of the SARS-CoV-2 virus, as well as the decline in antibody levels that are evident from studies which have shown a reduction in anti-S antibodies three weeks after the second dose of vaccination[26]. These findings provide a good rationale for the

administration of booster doses in preventing severe outcomes of COVID-19, although the primary vaccine series still appears effective in reducing mortality.

Our study has several strengths. First, the study included a large, comprehensive data set, which covers the entire population of Abu Dhabi, the largest emirate of the UAE. Second, the data was matched for age, sex, comorbidities, ethnicity, and the month of observation, in order to standardize the comparison and to avoid potential confounding factors. Third, the data we used provided a long follow-up period reaching almost 12 months. These allowed us to estimate, with great precision, the overall effectiveness, the effectiveness across different risk factors and the waning of effectiveness over time. Furthermore, we measured vaccine effectiveness according to three-month intervals (October–December 2020, January–April 2021, and May–July 2021). In each interval participants were follow-up for a maximum of three months. This allowed us to control for waning of effectiveness over time and to study the impact of variants of concern on effectiveness. This was possible because each three-month interval did correspond to specific predominant variants (the wild-type virus was predominant in October–December 2020, the alpha variant was predominant in January–April 2021 and delta was predominant in May–July 2021) [Unpublished data for the Abu Dhabi Public Health Center, Department of Health, Abu Dhabi, UAE].

Limitations of our study include looking only at hospitalization, critical care admission, and death as outcomes. We did not estimate the effectiveness of the vaccine against other outcomes, such as symptomatic infections or organ injury. Furthermore, we adjusted for factors, such as age, sex, ethnicity, comorbidities, and the month of entry into the study, but we did not account for other factors that may have influenced the outcomes such as obesity, smoking and occupation. Another limitation of our study is though the decline in vaccine effectiveness over time were analyzed we could not rule out the effect of the changes in the circulating variants on the waning of vaccine effectiveness over months. Finally, we did not include infections due to COVID-19 as one of the outcomes, primarily because the UAE has one of the highest per capita screening programs for COVID-19 worldwide. For public health relevance, we primarily investigated the severe and hospitalized COVID-19 cases, which are a better indicator of vaccine effectiveness.

We would like to conclude that the inactivated BBIBP-CorV (Sinopharm) vaccine was effective in preventing and reducing COVID-19 related hospitalizations and critical care admissions, as well as mortality. These findings also provide insights on the impact of risk factors on vaccine effectiveness. Our data further confirm the need for booster doses to increase protection against severe COVID-19 outcomes and highlight the necessity of continued monitoring of vaccine effectiveness over time to inform policy.

## Methods

**Study population and design.** We conducted a retrospective cohort study of individuals 18 years of age or older residing in Abu Dhabi, the capital emirate of the UAE. Clinical, demographic, and vaccination data of all subjects were extracted from the Abu Dhabi Health Services Company (SEHA) electronic health database and the Abu Dhabi health information exchange platform (Malaffi), which is a centralized electronic database of patient health information. SEHA is Abu Dhabi's largest health services provider, which operates all public hospitals and clinics in the emirate of Abu Dhabi. SEHA provides vaccination against COVID-19 to all residents of the UAE, manages vaccine-related complications, and operates all COVID-19 dedicated hospitals in Abu Dhabi.

The data on variables of interest was extracted retrospectively from the electronic medical records using Cerner Millennium software (Version 3.30), for both vaccinated and unvaccinated groups. All data within Cerner Millennium was recorded by clinicians during episodes of care, validated and signed/co-signed. The data extraction process was performed by two clinical analysts using the Discern Analytics 2.0 module in Cerner Millennium, in which the Cerner Command Language (CCL) was used to execute commands and retrieve the data. Five sets of raw data with different variables were extracted to Microsoft Excel 16.0. These extracts were identified by the clinical analysts based on the availability of data elements required to proceed with the analysis: Hospitalization, Risk Factors, Mortality, Critical Care units' admissions and Vaccination status. Manual validation steps against the electronic medical charts were conducted to ensure the data quality and a unique patient identifier used to link the five data sets in Power Bi Desktop (2.95.804.0) to produce the final spreadsheet that was analyzed.

The study population included both male and female individuals who are 18 years of age and above, residing in the UAE, vaccinated with BBIBP-CorV (Sinopharm) vaccine or not having received any vaccination against COVID-19.

This study was approved by the Institutional Review Board of the Department of Health, Abu Dhabi. Approval number: DOH/CVDC/2021/658. The review board waived the requirement for individual informed consent. All investigators had access to only anonymized patient information.

**Outcomes and covariates.** Vaccine effectiveness was investigated for three COVID-19 outcomes: hospitalization due to COVID-19, admission to the critical care unit due to COVID-19, and COVID-19–related death. Each study participant was followed for a period of three months from the date of entry (baseline date) into the study. The baseline date is defined as 14 days after the receipt of the second dose of vaccination for a vaccinated individual and his/her corresponding matched unvaccinated control. Vaccine effectiveness estimation was adjusted for several covariates, including age, sex, comorbidity, ethnicity, and the calendar month of entry into the study. The calendar month was divided into three groups for analysis from October 2020 to December 2020, January 2021 to April 2021, and May 2021 to July 2021. The calendar month of entry into the study was considered to be a critical factor and was adjusted for in our analysis, as the hazard varied over time due to changes in the rate of vaccination, newer treatment interventions, and the emergence of various SARS-CoV-2 variants of concern over the study period from October 2020 to September 2021.

**Statistical analysis.** Baseline characteristics were summarized using descriptive statistics, including mean and standard deviation (SD) for continuous measures and frequency tables for categorical variables. We compared categorical variables using the Chi-squared or Fisher's exact tests, and continuous variables using the unpaired $t$ test or its non-parametric equivalent (Wilcoxon rank sum test) in case the normality assumption was violated.

**The matching procedure.** Each vaccinated individual was matched to an unvaccinated control in 1:1 ratio using the Rolling Entry Matching (REM)[27]. Vaccinated subjects represent all individuals who completed their second dose of the vaccine prior to July 1st, 2021, and unvaccinated subjects were defined as individuals who did not receive any dose of the COVID-19 vaccination prior to July 1st, 2021. Vaccinated and unvaccinated individuals were matched according to their age, sex, ethnicity, and comorbidities, including diabetes, asthma, hypertension, chronic kidney diseases, cardiovascular diseases, immunodeficiency, transplant history, neoplasm and chronic respiratory diseases (other than asthma). The REM matching procedure matched each vaccinated individual to an unvaccinated individual according to the date of entry into the study (defined as 14 days post the date of the second dose for vaccinated individuals) to ensure that both vaccinated and unvaccinated individuals had the same exposure duration. The roll matching procedure uses exact matching for the date of entry into the study and the propensity matching score for all other covariates. All individuals were followed from their date of entry into the study up to September 30th, 2021.

**Estimation of the effectiveness against COVID-19 outcomes.** Vaccine effectiveness against each of the three outcomes was assessed using a Cox-proportional hazard model with time varying coefficients to consider the proportional hazard assumption and consequently the decline in effectiveness over time. For each of the three pre-determined COVID-19 outcomes (hospital admission, critical care admission or death), the time to event was defined as the duration of time (in days) from the baseline date until the date of occurrence of the outcome. Participants who did not experience the outcome during the follow-up period, or who were free from the outcome at the end of the follow-up period, were considered as right-censored.

Vaccinated individuals with time interval between 1st and 2nd vaccine dose less than two weeks were excluded from this analysis, along with those who had a history of COVID-19 hospitalization prior to their baseline date. Individuals who were matched to these exclusion sets were also removed to preserve the balance in baseline characteristics produced by the matching procedure. Only participants with no history of COVID-19 outcomes prior to the baseline date were considered in this survival analysis. In the primary analysis, unvaccinated participants who were administered at least one dose of the vaccine post-baseline (i.e., during their follow-up period) were excluded, along with their matched vaccinated individuals. In a sensitivity analysis, these individuals were re-included and censored at the date of their first dose. Their corresponding matched vaccinated individuals were also censored at the same date if they did not experience an outcome prior to that date.

Vaccine effectiveness was derived from the Cox proportional hazards model and was calculated as (1-Hazard Ratio) expressed as a percentage. Variables adjusted for in this model were age, sex, comorbidity, ethnicity, and the calendar month of entry into the study. The vaccine effectiveness, stratified by each of the risk factors listed above, was estimated by adding an interaction term into the survival model between the vaccination status and each of the risk factors. This subgroup analysis improves our understanding of the difference in vaccine effectiveness according to age groups, sex, ethnicity, associated comorbidities and the calendar month of entry.

The vaccine effectiveness, and its corresponding association with risk factors, was primarily assessed at three months after complete vaccination. However, to estimate the decline in vaccine effectiveness over time against each of the COVID-19 outcomes, the vaccine effectiveness was estimated at monthly intervals for follow-up periods extending beyond three months to nearly twelve months (after baseline). Effectiveness for each follow-up period was estimated using the Cox proportional hazard model and graphically plotted to show the trend over time.

Overall survival curves for the three COVID-19 outcomes were estimated and plotted using Kaplan–Meier approach[28]. We also estimated the absolute risk and the incidence rate of COVID-19 outcomes per 1000-person days according to vaccination status. All statistical analyses were performed using R software version 4.1.0[29]. $P < 0.05$ were considered statistically significant.

**Reporting summary**. Further information on research design is available in the Nature Research Reporting Summary linked to this article.

## Data availability
According to Abu Dhabi Department of Health regulations, individual-level data cannot be shared openly. Specific requests for remote access to de-identified data should be referred to DOH, Research committee. Requests sent to IRB, DOH medical research and development division (medical.research@doh.gov.ae) will be considered within 21 days pending IRB approval and DOH regulations. Auxiliary and summary data generated from the analyses are available in the supplementary file.

## Code availability
The code used for this study has been deposited in the git repository (https://github.com/abderrahimoulhaj/vaccine_effectiveness_nature_communications.git).

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

## Acknowledgements
We acknowledge the contribution of all individuals who helped in data collection and management.

## Author contributions
N.A.K., A.O., W.E., W.Z. contributed to the study concept and design; J.A., A.R., O.N., A.A., A.K., A.sA., B.S., H.S., S.H., J.S.A., D.M., A.H., M.A., S.W.,S.E., N.K., J.N.A., S.G.H., R.B. contributed to data collection; A.O. contributed to the statistical analysis; A.O., J.N.A., A.R., S.E., H.K. contributed to data management; A.O., S.G. contributed to data visualization; A.O., S.G. contributed to the original draft; N.A.K., A.O., S.G., A.A., H.I., H.S. revised the manuscript. N.A.K., A.O., S.G., A.A., H.I., H.S., F.H., O.N., J.A., A.K., A.sA., B.S., S.H., J.A., D.M., A.H., M.A., A.R., S.W., I.E., N.K., J.A., S.G.H., R.B., W.E., M.M., R.H., W.Z. critically reviewed and edited the final draft of the manuscript. All authors approved the final version of the manuscript.

## Competing interests
The authors declare no competing interests.

**Additional information**

