## [Peer Review File · Nature Communications]

Effectiveness of BBIBP-CorV vaccine against severe outcomes of COVID-19 in Abu Dhabi, United Arab EmiratesREVIEWER COMMENTS

Reviewer #1 (Remarks to the Author):

Overall Comment

This study is a large, carefully done, study of the effectiveness of the BBIBP-CorV vaccine in Abu Dhabi. The study compared vaccinated individuals to matched controls, matched on many important covariates including calendar time of entering into the study. The study provides a useful look into the effectiveness of this vaccine.

Specific Comments

1. (literature review, and line 214) There have been several studies on the effectiveness of the BBIBP-CorV vaccine (see <https://view-hub.org/covid-19/effectiveness-studies> for references). These should be mentioned and discussed in the introduction. For example, how does the effectiveness of BBIBP-CorV from your study compare to that done in other studies? Also, your statement about this being the first study on this topic on line 214 should be updated or deleted.
2. Names for the vaccine: It would be helpful to list other names for the BBIBP-CorV vaccine that have been used in the literature. For example, in reference 7, the vaccine with 78% vaccine efficacy is called an inactivated vaccine developed from SARS-CoV-2 HB02. Is that the same as BBIBP-CorV? Is CoronaVac by Sinovac the same vaccine?
3. (line 135) This sentence is confusing because Efron's approach to account for tied data is used with the Cox proportional hazards model, not with the Kaplan-Meier estimator.
4. (line 143) Please state whether or not unvaccinated controls were checked to make sure they were not previously hospitalized due to COVID-19.
5. (line 152) You state: "The time to event in both groups were censored at the date of receiving the vaccine post-baseline, resulting in a total of 1,111,934 vaccinated individuals and 1,111,934 unvaccinated matched controls." But censoring is a statistical term for partially observing follow up until the individual is not at risk anymore (because of for example, getting vaccine). At line 114, it is stated that unvaccinated participants that got vaccine post-baseline were excluded. Were those vaccinated post-baseline censored or excluded? I think they were excluded because of the matching sample size numbers you gave (i.e., the 1,111,934 in each of the two groups).
6. (line 220) You state that your finding were lower than those reported in the Phase III trial, but the vaccine efficacy reported in the abstract of that paper were both less than 79%. The only vaccine efficacy values that were greater in the Phase III trial was for the incident severe cases (see Table 2, ref 7).
7. (line 225) I think reference [15] is incorrect there, since that is about the REM matching.
8. (line 261) You state: "the vaccine effectiveness reduced during the months of January to April 2021 and this could be attributed to the rise in the number of COVID-19 cases..." But vaccine effectiveness should be mostly unaffected by the baseline rate, since it is defined by comparing the control group to the vaccinated group.
9. (line 272) When mentioning the vaccine efficacy of other studies, please say which vaccine you are talking about.
10. (line 273) I think reference [12] is incorrect, because the title suggests it is not a study in the United States.
11. (line 275) In the discussion section, please address the possibility that the define in effectiveness over time is related to variants of concern. A higher percentage of the study population at risk at 12 months would be during later calendar time when the variants of concern

were circulating more.

12. (lines 288-299) Perhaps talk about the previous point as a limitation of the study (but not only this study, but almost all studies of COVID-19). In other words, it is hard to tease out waning vaccine efficacy from changes in the circulating variants.

13. (Table S1) My version of Table S1 is difficult to read because many of the results are split into two rows. For example, the total n of 2,223,868 is written as "2,223,86" on one line and continues on the next line as "8".

Reviewer #2 (Remarks to the Author):

This is an important paper and the authors are to be commended for doing this important study. The information can be made more helpful by responding to the comments below.

First, the authors should add a few more sentences about the vaccine (presumably it is a whole virus that is inactivated). Information should be included on how the vaccine is made (or a reference to that) and definitely in this article, the recommended dosing interval should be mentioned.

Second, at the time of this study were any other vaccines in use in Abu Dhabi? If so, how were persons who received other vaccines handled?

Third, the authors mention 3 major 3 month periods to estimate decline of VE over time. There are two major reasons VE can decline: 1) waning immunity and 2) viral evasion of vaccine-induced immunity through antigenic changes. Can the authors add to Table 2 a column for the predominant SARS-CoV-2 strain circulating during each of the 3 periods?

Fourth, also on Table 2, the 4th row dealing with comorbidities has in the 9% CI column related to death, a lower bound of -8809.4. Is there an explanation for this?

Fifth, Table 1 compares the characteristics of vaccinees versus non-vaccinees. This reviewer perceives there were no statistically significant differences. If so, please add a footnote to the table saying that. If there are any statistically significant differences, that should be highlighted.

Sixth, the reasons for higher VE in persons with co-morbidities is discussed on page 11 of the manuscript. It is not clear how some of the issues raised explain this higher VE. For example, if persons with co-morbidities were vaccinated earlier, why would that increase VE since they were matched with persons also with co-morbidities. This issue should be clarified.

Figure 3 is hard to read in black and white. It is hard to see which line refers to each outcome. Similar problems are seen with Figure 2. This reviewer recommends the figures be modified to make the differences by category easy to see.

Response to Reviewers Comments

We express our gratitude to the reviewers for taking time to review our manuscript and providing valuable feedback that has significantly improved our manuscript. We have endeavored to revise our manuscript thoroughly in line with reviewers' suggestions. To this end, we have provided detailed responses to each comment raised by the reviewers. We have also incorporated the reviewers' suggestions and submitted the revised manuscript as cleaned and track changed versions.

Reviewer 1 comments	Response	Changes made
1. (literature review, and line 214) There have been several studies on the effectiveness of the BBIBP-CorV vaccine (see https://view-hub.org/covid-19/effectiveness-studies for references). These should be mentioned and discussed in the introduction. For example, how does the effectiveness of BBIBP-CorV from your study compare to that done in other studies? Also, your statement about this being the first study on this topic on line 214 should be updated or deleted.	We updated the literature review and included the newer studies on the effectiveness of the BBIBP-CorV vaccine into the Introduction and Discussion sections. We have deleted line 214.	Page 3 Lines- 59-64 Page 14 Lines – 292-295 Page 17 Lines 367-369
2. Names for the vaccine: It would be helpful to list other names for the BBIBP-CorV vaccine that have been used in the literature. For example, in reference 7, the vaccine with 78% vaccine efficacy is called an inactivated vaccine developed from SARS-CoV-2 HB02. Is that the same as BBIBP-CorV? Is CoronaVac by Sinovac the same vaccine?	We apologize for the confusion. We now include all vaccine names in the Introduction. We have also clarified that inactivated BBIBP-CorV vaccine and the inactivated vaccine developed from SARS-CoV-2 HB02 are the same and branded as the Sinopharm vaccine.	Page 2 Lines-39-41
3. (line 135) This sentence is confusing because Efron's approach to account for tied data is used with the Cox proportional hazards model, not with the Kaplan-Meier estimator.	We agree with the reviewer. The Efron method is used for the semi-parametric Cox proportional hazard model, and not for the Kaplan Meir. The sentence is now updated and the word correction for ties is removed.	Page 8 line 172-173
4. (line 143) Please state whether or not unvaccinated controls were checked to make sure they were not previously hospitalized due to COVID-19.	We apologize for this omission. We now have removed all individuals previously hospitalized with COVID-19 (representing 5451 Vaccinated and 6779 Unvaccinated) along with their corresponding matched individuals. This resulted in a total of 24460 individuals excluded from the primary analysis. The flowchart has been updated accordingly.	Page 9 Lines 190-193 Page 11 Lines 226-228 Flowchart 1 Supplementary table (eTable 1.2)

	We found out that after removing individuals with past history of COVID-19 hospitalization, the effectiveness against the three outcomes has remained almost the same. Furthermore, in the previous analysis the variable age was used in the Cox PH model with three categories <40, [40,60] and >60. These categories have now been reduced to <60 and >=60. The reason for this regrouping is that our data revealed no deaths for individuals with age <40, which affects the estimates of the hazard ratio as the coefficients are in the border of the parameter space. The same is done for the variable ethnicity now grouped as “Arab” versus “Others”. These changes do not affect the previously estimated effectiveness. We have now updated the supplementary material and included more results from both primary and sensitivity analyses.	
5. (line 152) You state: “The time to event in both groups were censored at the date of receiving the vaccine post-baseline, resulting in a total of 1,111,934 vaccinated individuals and 1,111,934 unvaccinated matched controls.” But censoring is a statistical term for partially observing follow up until the individual is not at risk anymore (because of for example, getting vaccine). At line 114, it is stated that unvaccinated participants that got vaccine post-baseline were excluded. Were those vaccinated post-baseline censored or excluded? I think they were excluded because of the matching sample size numbers you gave (i.e., the 1,111,934 in each of the two groups).	Thank you for pointing out to this detail. The paragraph has been updated accordingly. Indeed, as the reviewer stated, in the primary analysis, we exclude unvaccinated controls (along with their matched vaccinated individuals) who were vaccinated post-baseline. In a sensitivity analysis, however, these individuals were re-included and censored at the date of their first dose. Their corresponding matched vaccinated subjects were also censored at the same date if they did not experience an outcome prior to that date (see updated flowchart)	Page 9-10 Lines 194-202
6. (line 220) You state that your finding were lower than those reported in the Phase III trial, but the vaccine efficacy reported in the abstract of that paper were both less than 79%. The only vaccine efficacy values that were greater in the Phase III trial was for the incident severe cases (see Table 2, ref 7).	The phase III trial calculated the efficacy against symptomatic infections. Our study did not include infections as one of the outcomes. The phase III trial showed the efficacy was 100% against severe cases in a secondary analysis (based on 2 deaths in Placebo and 0 deaths in the vaccine arm). In this real world study, we evaluated the effectiveness against severe outcomes and found the effectiveness was lower than the percentage reported in	Page 14 Lines 297-304

	phase III trial. We updated the manuscript according to the reviewer's comment	
7. (line 225) I think reference [15] is incorrect there, since that is about the REM matching.	We apologize for the error and have corrected the reference.	Page 6 Line -122
8. (line 261) You state: "the vaccine effectiveness reduced during the months of January to April 2021 and this could be attributed to the rise in the number of COVID-19 cases..." But vaccine effectiveness should be mostly unaffected by the baseline rate, since it is defined by comparing the control group to the vaccinated group.	We agree with the Reviewer that the statement was inaccurate. We have removed this argument and now state that the decline in vaccine effectiveness is attributed to the emergence of newer variants. During the months of January to July 2021 the predominant variants in the UAE were beta and delta.	Page 16-17 Lines 350-360
9. (line 272) When mentioning the vaccine efficacy of other studies, please say which vaccine you are talking about.	Vaccine names are now included.	Page 13, 15 and 17 Lines – 284, 323 and 371
10. (line 273) I think reference [12] is incorrect, because the title suggests it is not a study in the United States.	We apologize for this error. This reference has been changed. We reviewed and updated all references.	Page 17 Lines - 369
11. (line 275) In the discussion section, please address the possibility that the decline in effectiveness over time is related to variants of concern. A higher percentage of the study population at risk at 12 months would be during later calendar time when the variants of concern were circulating more.	We agree with the Reviewer. The sentence now reads: "This decline in effectiveness over time is likely explained by the viral evasion of vaccine induced immune response through antigenic changes in newer variants of the SARS-COV-2 virus, as well as the decline in antibody levels that are evident from studies which have shown a reduction in anti-S antibodies three weeks after the second dose of vaccination."	Page 17 Lines 371-375
12. (lines 288-299) Perhaps talk about the previous point as a limitation of the study (but not only this study, but almost all studies of COVID-19). In other words, it is hard to tease out waning vaccine efficacy from changes in the circulating variants.	1- In this study, we measured vaccine effectiveness according to three-month intervals (Oct-Dec 2020, Jan-Apr 2021 and May-July 2021). In each interval participants were follow-up for a maximum of 3 months. This allows us to control for waning of effectiveness over time and to study the impact of variants on the effectiveness. This was possible because each three-month interval corresponded to specific predominant variants (Oct-Dec 2020: the most predominant variant was the wild virus, Jan-Apr 2021: the most predominant variant was Alfa, and May-	Page 18 Lines 386-396

	July 2021: the most predominant variant was Delta). 2- In the twelve-month follow-up, we looked at the decline over time in the effectiveness. In this particular analysis, we agree with the reviewer that it is hard to tease out waning vaccine efficacy from changes in the circulating variants. The effect of the changes in the circulating variants on the waning of vaccine effectiveness is included as a limitation, as suggested by the Reviewer.	
13. (Table S1) My version of Table S1 is difficult to read because many of the results are split into two rows. For example, the total n of 2,223,868 is written as “2,223,86” on one line and continues on the next line as “8”.	We have now updated the supplementary file to improve the clarity of all tables and figures. The supplementary file also now includes more results from the primary and sensitivity analyses.	Supplementary file

Reviewer 2 comments	Response	Changes made
First, the authors should add a few more sentences about the vaccine (presumably it is a whole virus that is inactivated). Information should be included on how the vaccine is made (or a reference to that) and definitely in this article, the recommended dosing interval should be mentioned.	As suggested by the Reviewer, details on vaccine development and dosing schedule are now included in the Introduction.	Page 2 Lines 40-45
Second, at the time of this study were any other vaccines in use in Abu Dhabi? If so, how were persons who received other vaccines handled?	During this study period, additional vaccines were made available to UAE residents. However, individuals who received vaccines other than Sinopharm were not included in this study analysis.	Page 4 Lines 84-86
Third, the authors mention 3 major 3 month periods to estimate decline of VE over time. There are two major reasons VE can decline: 1) waning immunity and 2) viral evasion of vaccine-induced immunity through antigenic changes. Can the authors add to Table 2 a column for the predominant SARS-CoV-2 strain circulating during each of the 3 periods?	We have now added a paragraph in the Discussion section on the decline in VE with regard to the predominant variants and decline over time. Moreover, the predominant strains identified during the study period are now added to table 2.	Page 17 Lines 371-377 Table 2

Fourth, also on Table 2, the 4th row dealing with comorbidities has in the 9% CI column related to death, a lower bound of -8809.4. Is there an explanation for this?	We agree with the Reviewer. The new results no longer show this lower value of -8809.4. One potential explanation for the previous results is because we were adjusting for the variable age with 0 deaths for those in the category <40. Similarly, in Nationality, there were 0 deaths in the others category. We now have regrouped age and nationality as binary variables and provide more results in the supplementary material for both the primary and sensitivity analyses. From another side, the value of -8809.4 is, however, possible for the lower band of the 95% of the effectiveness. This is because effectiveness is defined as $100 \times (1 - HR)$. As such, a negative value implies an estimated $HR > 1$.	Table 2 Supplementary table (eTable 1.2)
Fifth, Table 1 compares the characteristics of vaccinees versus non-vaccinees. This reviewer perceives there were no statistically significant differences. If so, please add a footnote to the table saying that. If there are any statistically significant differences, that should be highlighted.	We agree with the Reviewer. We checked the p-values comparing the distribution of baseline characteristics between vaccinated and unvaccinated in Table 1. The p-values are statistically significant, but of no significant clinical relevance. This is because the unpaired t-test, the Wilcoxon rank test or the Chi-square test will always show statistically significant differences when the sample size is quite large (over 1 million for each group) because of the very high power to detect even small differences. A lot of power means that we will detect significant differences that can be clinically meaningless in terms of very large sample	

	sizes. For instance, when using the average instead of the median in table 1, the average age in unvaccinated is 36.72 years compared to 36.74 years in Vaccinated. The p.value calculated was significant (P <0.001) despite there is no clinical difference.	
Sixth, the reasons for higher VE in persons with co-morbidities is discussed on page 11 of the manuscript. It is not clear how some of the issues raised explain this higher VE. For example, if persons with co-morbidities were vaccinated earlier, why would that increase VE since they were matched with persons also with co-morbidities. This issue should be clarified.	We agree with the reviewer. The previous explanation was not clear. The interpretation has now been updated in the main manuscript. We think that this difference in VE can be explained by the high risk of hospitalization among unvaccinated individuals with comorbidities, when compared to the significantly lower risk in individuals without comorbidities. As such, there was greater reduction after vaccination in patients with comorbidities compared to those without, and thereby increased vaccine effectiveness, in the population with comorbidities. We looked at the data again to understand this, and we tried to calculate the risk of hospitalization directly from the data without using any statistical model and without any adjustment. We found out the following: Patients without comorbidity:  - The risk of hospitalization among unvaccinated = 0.0006 - The risk of hospitalization among vaccinated = 0.0002 	Page 15-16 Lines 331-340

	This leads to a crude VE of: $VE = 1 - (0.0002/0.0006)$ $= 67\%$ Patients with comorbidity:  - The risk of hospitalization among unvaccinated = 0.0291 - The risk of hospitalization among vaccinated = 0.0035 This leads to a crude VE of: $VE = 1 - (0.0035/0.0291)$ $= 88\%$	
Figure 3 is hard to read in black and white. It is hard to see which line refers to each outcome. Similar problems are seen with Figure 2. This reviewer recommends the figures be modified to make the differences by category easy to see.	Figure 2 and 3 have now been updated accordingly.	Figure 2 and Figure 3

REVIEWERS' COMMENTS

Reviewer #1 (Remarks to the Author):

1. (lines 78-79) This statement is not accurate, since there are unvaccinated controls included in this study.

2. (lines 254-255) You state: "Our findings showed lower effectiveness than those reported in the secondary analysis of the phase III trial, which showed 100% efficacy against COVID-19-related deaths. [8]" However, when I looked up that paper, they had no mortality during follow-up in any arm, and only 2 severe cases in the control arm. So the 100% efficacy in that trial is against severe cases or death, not against death. Further, the 100% vaccine efficacy estimate against severe disease/death was based on 2 cases versus 0 in the two vaccine arms, and it would have a very small lower 95% confidence interval limit if it was calculated (but it was not calculated). So there is no good vaccine efficacy estimate for death from the randomized trial (since it is 0 deaths in all three arms).

3. (line 278) You state: "No sex differences in mortality were found." Please state "No significant sex differences..." instead, since you observed VE=77.0% for female, and VE=88.9% for males.

4. In the response to my (Reviewer 1) comment 12, you gave two responses. The second one stated: "In the twelve-month follow-up, we looked at the decline over time in the effectiveness. In this particular analysis, we agree with the reviewer that it is hard to tease out waning vaccine efficacy from changes in the circulating variants. The effect of the changes in the circulating variants on the waning of vaccine effectiveness is included as a limitation, as suggested by the Reviewer." I could not find any mention in the limitations paragraph of the changes in the circulating variants being a limitation of your study for estimating the long term vaccine efficacy. Did you forget to put that into the limitations discussion?

Reviewer #2 (Remarks to the Author):

The authors have well replied to this reviewer's prior comments.

This reviewer has just a few minor comments.

On page 10, starting on line 215, the authors mention the vaccine effectiveness (VE) in individuals for which the VE is higher. While the key information is readily available in Table 2, it still would be helpful to the reader if the comparator VE (e.g., males compared to females) was also in the text. And on line 219, "Table 2" should be cited.

Figure 3 is hard to read in black and white. If the publication will be in black and white, this reviewer recommends using different lines such as solid, dotted, and dashed for the 3 outcomes.

Response to Reviewer Comments

Reviewer Comments Reviewer #1	Response to comments	Changes made in track changes
1. (lines 78-79) This statement is not accurate, since there are unvaccinated controls included in this study.	The statement has been modified and now reads: The study population included both male and female individuals who are 18 years of age and above, residing in the UAE, vaccinated with BBIBP-CorV (Sinopharm) vaccine or not having received any vaccination against COVID-19.	Lines 409-411
2. (lines 254-255) You state: “Our findings showed lower effectiveness than those reported in the secondary analysis of the phase III trial, which showed 100% efficacy against COVID-19-related deaths. [8]” However, when I looked up that paper, they had no mortality during follow-up in any arm, and only 2 severe cases in the control arm. So the 100% efficacy in that trial is against severe cases or death, not against death. Further, the 100% vaccine efficacy estimate against severe disease/death was based on 2 cases versus 0 in the two vaccine arms, and it would have a very small lower 95% confidence interval limit if it was calculated (but it was not calculated). So there is no good vaccine efficacy estimate for death from the randomized trial (since it is 0 deaths in all three arms).	We have modified the statement as “Our findings showed lower effectiveness than those reported in the secondary analysis of the phase III trial, which showed 100% efficacy against severe COVID-19 cases, which included a composite of severe cases and death”	Line 295-296
3. (line 278) You state: “No sex differences in mortality were found.” Please state “No significant sex differences...” instead, since you observed VE=77.0% for female, and VE=88.9% for males.	The statement was modified as suggested. It now reads: No significant sex differences in mortality were found. “	Line 318
4. In the response to my (Reviewer 1) comment 12, you gave two responses. The second one stated: “In the twelve-month follow-up, we looked at the decline over time in the effectiveness. In this particular analysis, we agree with the reviewer	We apologize for this oversight. The limitations now include: “Also, although the decline in vaccine effectiveness over time was	Lines 373-375

that it is hard to tease out waning vaccine efficacy from changes in the circulating variants. The effect of the changes in the circulating variants on the waning of vaccine effectiveness is included as a limitation, as suggested by the Reviewer." I could not find any mention in the limitations paragraph of the changes in the circulating variants being a limitation of your study for estimating the long term vaccine efficacy. Did you forget to put that into the limitations discussion?	analyzed, changes in the circulating variants likely also impacted long-term vaccine efficacy."	
Reviewer #2 The authors have well replied to this reviewer's prior comments. This reviewer has just a few minor comments. On page 10, starting on line 215, the authors mention the vaccine effectiveness (VE) in individuals for which the VE is higher. While the key information is readily available in Table 2, it still would be helpful to the reader if the comparator VE (e.g., males compared to females) was also in the text. And on line 219, "Table 2" should be cited.	Thank you. The lines are modified and the comparators are now added as suggested by the Reviewer. Table 2 is now cited	Lines 249-261
Figure 3 is hard to read in black and white. If the publication will be in black and white, this reviewer recommends using different lines such as solid, dotted, and dashed for the 3 outcomes.	The new figure uploaded has a higher resolution in tiff. format The figures will be published as color images.